# Machine Learning Assisted Approach for Finding Novel High Activity Agonists of Human Ectopic Olfactory Receptors

**DOI:** 10.3390/ijms222111546

**Published:** 2021-10-26

**Authors:** Amara Jabeen, Claire A. de March, Hiroaki Matsunami, Shoba Ranganathan

**Affiliations:** 1Applied BioSciences, Macquarie University, Sydney, NSW 2109, Australia; amara.jabeen@mq.edu.au; 2Department of Molecular Genetics and Microbiology, Duke University School of Medicine, Durham, NC 27710, USA; claire.de.march@duke.edu; 3Department of Neurobiology, Duke Institute for Brain Sciences, Duke University, Durham, NC 27710, USA

**Keywords:** machine learning, random forest, molecular descriptors, virtual ligand screening, olfactory receptor, G protein-coupled receptors, luciferase assay

## Abstract

Olfactory receptors (ORs) constitute the largest superfamily of G protein-coupled receptors (GPCRs). ORs are involved in sensing odorants as well as in other ectopic roles in non-nasal tissues. Matching of an enormous number of the olfactory stimulation repertoire to its counterpart OR through machine learning (ML) will enable understanding of olfactory system, receptor characterization, and exploitation of their therapeutic potential. In the current study, we have selected two broadly tuned ectopic human OR proteins, OR1A1 and OR2W1, for expanding their known chemical space by using molecular descriptors. We present a scheme for selecting the optimal features required to train an ML-based model, based on which we selected the random forest (RF) as the best performer. High activity agonist prediction involved screening five databases comprising ~23 M compounds, using the trained RF classifier. To evaluate the effectiveness of the machine learning based virtual screening and check receptor binding site compatibility, we used docking of the top target ligands to carefully develop receptor model structures. Finally, experimental validation of selected compounds with significant docking scores through in vitro assays revealed two high activity novel agonists for OR1A1 and one for OR2W1.

## 1. Introduction

G protein-coupled receptors (GPCRs, also known as seven transmembrane or 7TM receptors) represent the largest family of cell surface receptors. The myriad functional diversity of GPCRs has led them to be the largest family of proteins targeted by approved drugs. Primarily the drugs that target the GPCRs are small molecules and peptides [1]. Olfactory receptors (ORs) [2], first reported in 1991, represent the largest sub-group of G protein-coupled receptors (GPCRs) [3].

Initially, ORs were thought to be localised only to nasal tissue and responsible solely for the sense of olfaction, with odorant molecules combinatorically leading to the perception of smell [4]. However, some ORs are expressed in extra-nasal tissues such as mammalian germ cells [5], where they are implicated in different physiological and disease conditions. A recent study has reported the localization of a subset of ORs in various tissues including the brain, prostate, sperm, colon, breast, lungs and kidneys [6]. Functional characterization of these ectopic ORs in different tissues support their roles in cell-cell recognition, migration, proliferation, apoptosis, exocytosis, and novel alternate pathways. Ectopic ORs are also known to be associated with numerous diseases and disorders, including prostate cancer, melanoma, colon cancer, breast tumours, neurodegenerative disorders, obesity and anaemia [7]. Thus, ORs are potential therapeutic targets [8]. Recently, del Marmol et al., reported the experimental structure of an insect olfactory receptor through cryo-electron microscopy [9], which has an inverted topology to animal ORs. However, there is no experimental structure for any animal OR. We have reviewed the challenges associated with the experimental structure determination for ORs elsewhere in detail [10]. Briefly, the absence of any experimentally determined animal OR structure is attributed to ORs being low abundance, tissue-specific hydrophobic membrane proteins, which are difficult to crystallize. Further, ORs show poor trafficking to the plasma membrane, due to mRNA retention when expressed heterologously in different cell types [11].

Linking the olfactory stimulus repertoire, consisting of more than a trillion, to its counterpart ORs, is a challenging task. To date, only 21% of human ORs have been matched (or deorphanized) with active ligands [10]. OR deorphanization using olfactory sensory neurons (OSNs) can be vastly facilitated by computational approaches. In silico methods coupled with in vitro approaches have proven useful in deorphanizing some ORs [12]. Recent excellent studies using pharmacophore based virtual screening [13] and machine learning (ML) [14] have resulted in expanding the chemical space of a few ORs including the prostate specific G protein receptors (PSGRs: OR51E1 and OR51E2). Although there are a wide variety of ML algorithms, no single algorithm has been capable of solving every problem [15]. ML has now been extensively used to solve various bioinformatics problems [16,17], including GPCR research [18,19]. Recently, support vector machines (SVMs) were used to predict agonists for OR51E1, OR1A1, OR2W1 and MOR256-3 from commonly used odorants [14]. 7/18 predicted ligands for OR1A1, 2/5 for OR2W1, 5/13 for MOR256-3 and 2/4 for OR51E1 were found to be the true ligands when verified through in vitro luciferase assays. The OR51E1 homology model was then used in the reported study to elaborate binding cavity, mutating the residues predicted by molecular docking resulted in receptor response termination in vitro. Homology modelling and molecular docking are thus powerful tools to study receptor ligand interactions in the absence of experimental 3D structure. Many studies have been reported that couple homology modelling, molecular docking and site-directed mutagenesis to elucidate the binding cradle of different ORs with various odorants [20,21,22,23,24]. The mutational dataset for human ORs is now available through an interactive webserver, the Human Olfactory Receptor Mutation Database (hORMdb) [25]. Virtual screening using homology models has also resulted in the discovery of novel ligands. Recently, the human metabolome database was screened against the homology model of OR51E2 and resulted in identification of 24 novel agonists and one antagonist verified experimentally. In a benchmarking study [26], homology models of 19 GPCRs were used for ligand based virtual screening and 10 models showed comparable performance to X-ray structures depicting the applicability of homology models for the identification of novel ligands. We recently compared the performance of four classifiers, based on agonist and non-agonist datasets for OR1G1, with the naïve Bayes classifier performing better than SVM, random forest [27] and neural networks (NN) for agonist prediction [28].

Tunyasuvunakool et al., have reported highly accurate protein structure prediction for the entire human proteome, including ORs [29]. However, the AlphaFold models generated in this study need a lot of adjustment in the orientation of the transmembrane helices (Jabeen and Ranganathan, unpublished data), to recover the OR binding sites that were validated by published mutational studies [20,21,22,23,24]. Instead, we recently showed that using a biophysical approach, Bio-GATS, for template selection generates an excellent homology model for OR1A1 [30].

In the current study, we have focused our attention on two ectopic ORs with broad ligand spectrum, OR1A1 and OR2W1. OR1A1, reported in gut enterochromaffin cells, was implicated in serotonin release [31]. Recurrent mutations in OR1A1 were identified in lung adenocarcinoma [32]. Further, OR1A1 was detected in HepG2 liver cells [33], and implicated in hepatic triglyceride metabolism modulation. OR2W1 has recurrent mutations reported in small lung cancer [34]. Both receptors, along with a few other ORs, are also proposed to have roles in fatigue attenuation [35].

The aim of our study was to expand the chemical space for these two ORs with potential clinical importance, by predicting and experimentally testing novel agonists. Therefore, we have developed an ML-based workflow (Figure 1) to predict agonists for the two ORs by scanning the huge chemical compounds databases available online. We selected three methods, RF, SVM and NB, based on their performance for OR1G1 [28]. We filtered the predicted compounds using knowledge-based homology models of the two receptors, based on the best Bio-GATS template [30]. From the shortlisted predictions, we validated some of the randomly selected predicted compounds using an in vitro functional assay.

## 2. Methods

### 2.1. Ligand Dataset Collection for the Receptors and Chemical Descriptors Calculation

The experimentally tested compounds against the OR1A1 and OR2W1 were retrieved from the literature. Among the 365 compounds tested against OR1A1, 51 are agonists, 263 are non-agonists, and 51 have conflicting information (Appendix A). While for the OR2W1, 292 compounds have been experimentally tested, of which 64 are agonists, 198 are non-agonists, seven are antagonists, and 22 have conflicting information (Appendix A). As the antagonists are few in number and the compounds with conflicting information could not be classified uniquely, these two categories were not considered for further analysis. Molecular descriptors for the agonists and non-agonists of both the receptors were calculated using an open-source and free software, Mordred v1.2.0 [36]. Overall, 1443 and 1505 2D and 3D molecular descriptors were calculated for OR1A1 and OR2W1, respectively.

### 2.2. Data Pre-Processing, Feature Selection and Class Balancing

Since a large number of descriptors were calculated for both receptors, the pre-processing and feature selection techniques were employed to get an optimal number of features. Initially, all the features with any missing value were eliminated then a near zero filter was applied to exclude features having low variance. A correlation filter was applied to reduce the collinearity among the descriptors [37]. The threshold value for the correlation coefficient (*r*) was set to 0.95 as previously setup for OR1A1 and OR2W1 features [14] and for the moth odorant receptor [38]. Afterwards, three different methods were used for feature selection including a wrapper method: recursive feature elimination, a filter method: Gini, and an embedded method: random forest feature selection, were applied for selection of relevant subset of molecular descriptors. The dataset, comprising known agonists and non-agonists with the selected features, was split into 80% training and 20% test sets using random sampling. Pre-processing, feature selection and data splitting were carried out using the R programming language [39].

Since the two classes (agonists and non-agonists) are imbalanced (one agonist: ~five non-agonists for OR1A1 and one agonist: ~four non-agonists for OR2W1), we used the synthetic minority over-sampling technique (SMOTE) [40] embedded as a node in Knime 3.6.0 [41] on the training-set to have balanced datasets. The 5th nearest neighbour was considered for synthetic sampling.

### 2.3. Classifiers

We generated RF, SVM, and NB classifier models using the R programming language. 10-fold cross validation on the training dataset was used as a resampling method for each classifier. The CV was repeated three times to avoid any bias during the creation of CV data splits. A brief description of each model is provided below.

#### 2.3.1. Random Forest

The RF approach utilizes the decision trees and creates various models through random partitioning. The final output is based on majority voting [42]. In our model, the number of trees and variables randomly sampled as candidates at each split were hyper-parametrized to obtain the optimal RF model. The final RF classification model was based on 300 trees with 5 variables randomly sampled as candidates at each split for OR1A1 and 3 variables randomly sampled as candidates at each split for OR2W1.

#### 2.3.2. Support Vector Machine

SVM is based on calculating the maximal marginal hyperplane to separate positives from the negatives [43]. In the current study, the SVM classifier was built using radial basis kernel function. The two parameters that were hyper-parametrized are sigma and cost. Sigma was held constant at the value of 0.2326189 and the accuracy metric was used to select the optimal model using the largest value. The final value for sigma was 0.2326189 and 0.5 for cost.

#### 2.3.3. Naïve Bayes

NB is the commonly used, simple and computationally less expensive ML method [44]. NB is based on Bayes rule as mentioned in Equation (1):(1)P(y|x)=P(y)P(x|y)/P(x)
where y represents the class and x represents the data points.

The NB classifier assumes that all features are independent of each other so P(y) P(x|y) can be re-written as Equation (2):(2)$$P\left(y_{j}\right)P\left({x|y}_{j}\right)=P\left(y_{j}\right)\overset{n}{\underset{i=1}{\prod }}P(x_{i}|y_{j})$$
where P(y_j_) is the prior of the classes and P(x_i_|y_j_) is the distribution for one feature and one class. The Gaussian distribution was used in this study for NB classification model.

### 2.4. Model Validation

Prediction performance of each classifier was assessed by the test set for each OR and by 10-fold cross validation of the training data. 20% of the dataset was reserved as a test set and was not used for training the model. Therefore, this test set was unseen for the classifier and can be considered as a blind test set, as no suitable external validation set is available. Two statistical tests namely, *p*-value and Cohen’s kappa coefficient (*κ*) were also used to evaluate the models. Further, accuracy, sensitivity, specificity and the F1 score measures were used to evaluate the classifiers. The values for accuracy, sensitivity, and specificity were calculated using Equations (3)–(5):(3)Accuracy=TP+TNTP+TN+FP+FN
(4)Sensitivity/recall=TPTP+FN
(5)Specificity=TNTN+FP
where TP, TN, FP and FN refer to true positive, true negative, false positive, and false negative.

The F1 score is defined in Equation (6) as:(6)F1=Precision*recallPrecision+recall
where precision is calculated as Equation (7) and recall is calculated as Equation (4):(7)Precision=TPTP+FP.

### 2.5. Filtration of Compounds for Virtual Screening on the Basis of Chemical Similarity

For virtual screening using the built classifiers, we downloaded the compounds from ZINC [45], human metabolome database (HMDB) [46], ChEBI [47], Cancer Odor Database (COD) [48], OdorDB [49]. Those compounds that have already been experimentally tested against OR1A1 and OR2W1were filtered out from the list. We used PubChem fingerprints coupled with Tanimoto index for scanning similar spaced compounds from the above- mentioned databases. Only compounds with Tanimoto index of at least 85% were selected for screening. The final list of selected compounds was then evaluated as potential agonists of OR1A1 and OR2W1, using the RF classifier.

### 2.6. OR1A1 and OR2W1 Homology Modelling

The 3D models of OR1A1 (UniProtID: Q9P1Q5) and OR2W1 (UniProtID: Q9Y3N9) were built using homology modelling approach as described previously [50]. The X-ray crystal structure of bovine rhodopsin (PDB ID: 1U19) [51] was used as the template for homology modelling of human OR1A1 and OR2W1. Briefly, the sequences of OR1A1 and OR2W1 were aligned with bovine rhodopsin, based on conserved GPCR motifs (Appendix A). The predicted transmembrane domains in both receptors were based on the GRoSS sequence alignment of all known GPCRs sequences [52], as implemented by Bio-GATS [30]. Homology modelling was performed using MODELLER 9.18 [53]. The resulting models were assessed using the Modeller objective function, which reflects the quality of the model and the presence of a disulphide bond between Cys97 and Cys179. The selected models were also evaluated using the Ramachandran plot and favoured rotamers, on the Molprobity webserver [54]. The side chains of the built models were refined using SCWRL4 [55] to improve rotamer geometry.

### 2.7. Molecular Docking of Highly Probable Predicted Compounds

The compounds having similar space as agonists of OR1A1 and OR2W1 were classified as agonists and non-agonists through the trained RF classifier. Compounds with prediction probability of 1.0 for being agonists alone were considered for molecular docking. The binding pockets of both the receptors were predicted using ICMPocketFinder embedded in ICM package [56]. The binding pockets were selected based on site-directed mutagenesis data of different ORs. Induced fit docking was then performed using ICM. Ten conformations were generated for each predicted ligand and the control molecule. The docking effort was set to 3, as the developers of ICM benchmark the accuracy at this effort level. The conformation with the lowest ICM-score was selected for binding analysis.

### 2.8. Cell Culture

Hana3A cells [57] were maintained in minimal essential medium [34] containing 10% FBS (vol/vol) with penicillin-streptomycin and amphotericin B (1/200 each vol/vol) at 37 °C and 5% CO_2_. Hana3A cells are derived from HEK293T [57] and are optimized for OR studies as shown in several studies [58,59,60,61,62,63,64,65,66], compared to earlier OR expression in Sf9 insect cells in Gat et al. [67], and in *Xenopus* oocytes, COS-7, PC12h and CHO-K1 cells in Katada et al. [68].

### 2.9. Dual-Glo Luciferase Reporter Gene Assay

The Dual-Glo luciferase assay system (Promega, Madison, WI, USA) was used to evaluate the functionality of wild-type OR1A1 and OR2W1 in an in vitro system [69,70]. The open reading frames of ORs were amplified using Phusion polymerase (Thermo Fisher Scientific, Waltham, MA, USA). Amplified fragments were cloned into pCI expression vector (Promega, Madison, WI, USA) containing the sequence encoding the first 20 amino acids of human rhodopsin (Rho-tag) at N-terminal [71]. Hana3A cells have been cultured and plated the day before transfection with 6 mL at 1/10 of a 100% confluence 100 mm plate into 96-well plates coated with poly D-lysine. After overnight incubation, the required genes were transfected using, for each plate, 5 ng SV40-RL, 10 ng CRE-Luc, 5 ng human RTP1S [72], 2.5 ng M3 receptor [73] and 5 ng of receptor (OR1A1, OR2W1 or empty vector Rho-pCI) plasmid. After around 18 h of transfection, cells were stimulated during 3.5 h by 25 µL of odorant diluted in CD293 + 1% glutamine + 30 µM CuCl_2_. Odorants were obtained from Sigma Aldrich (St. Louis, MO, USA) and diluted at 1 M concentration in DMSO as stock solutions. Dose response curves were determined with concentrations of 0, 1, 3.16, 10, 31.6, 100, and 316 µM obtained by dilution of the DMSO stock solution in CD293 + 1% glutamine + 30 µM CuCl_2_. The luminescence of Firefly (Luc) and Renilla (Rluc) luciferase, were then sequentially monitored by injecting the corresponding substrate following the supplier’s protocol. The activity in each well was normalized as (Luc-400)/(Rluc-400). The response of the receptor was also normalized to its basal activity as (NL_X_/NL_0_)-1 where NL_0_ is the normalized luminescence value at 0 µM of odorant and NL_X_ the value at X µM. The cell response upon odorant stimulation was attributed to an OR if the empty vector control showed no response, assuring that the cell response is not due to other parameters than the presence of the OR at the cell surface. Raw results were first analyzed with Excel (Microsoft Corporation, Albuquerque, NM, USA) and dose response curves, max efficacy and EC50 have been determined with GraphPrism 6 software (GraphPad Software, La Jolla, CA, USA). Areas under the curves (AUC) were calculated in Excel by summing all the OR responses at different concentrations for each odorant.

### 2.10. Cell Surface Expression Evaluation by Flow Cytometry

The cell surface expression of the studied ORs has been evaluated by flow cytometry, which provides more quantitative cell surface expression data than conventional immunostaining, using the following protocol [74]. Human embryonic kidney variant 293T (HEK293T) cells were grown to confluency, resuspended and seeded onto 35 mm plates at 25% confluency. The cells were cultured overnight. The OR, RTP1s and GFP were transfected using Lipofectamine 2000. After 18–24 h, the cells were resuspended by cell stripper and then kept in 5 mL round bottom polystyrene (PS) tubes (Falcon 2052, Corning, Corning, NY, USA) on ice. The cells were spun down at 4 °C and resuspended in phosphate-buffered saline (PBS) containing 15 mM NaN_3_, and 2% foetal bovine serum (FBS) to wash the cell stripper. They were incubated in ice with primary antibody (mouse anti-Rho4D2 [75]) and then washed, and stained with phycoerythrin (PE)-conjugated donkey anti-mouse antibody (Jackson Immunologicals, West Grove, PA, USA) in the dark. To stain dead cells, 7-aminoactinomycin D (Calbiochem, MilliporeSigma, Burlington, MA, USA) was added. The cells were analyzed using FACS (BD FACSCanto II, Bio-Rad Laboratories, Hercules, CA, USA) with gating, allowing for GFP positive, single, spherical, viable cells, and the measured PE fluorescence intensities were analyzed and visualized using Flowjo v10.0.8 [76]. We also added Olfr539, which is robustly expressed on the cell surface, and Olfr541, which shows no detectable cell surface expression, as positive and negative controls of OR cell surface expression [74], respectively.

## 3. Results

### 3.1. Chemical Diversity Analysis

The chemical space for OR1A1 and OR2W1 is highly diverse and comprised of aldehydes, alcohols and esters among others (Appendix A). The diversified nature of the collected experimentally known compounds against the two receptors were verified using principal component analysis (PCA). The first two principal components were plotted and are clearly indicative of the diversified chemical space for the two receptors (Figure 2A). Splitting the experimentally tested data into 80% training and 20% test datasets showed considerable overlap between the two sets (Figure 2B) indicating that the classifiers are being validated on the basis of similar spaced compounds.

### 3.2. Feature Selection and Performance of the Classifiers

The strategy for identifying novel agonists for OR1A1 and OR2W1 required training of the classifiers with the chemical descriptors (or features) of the known agonists and non-agonists. Mordred calculated 1505 for OR1A1 and 1443 features for OR2W1. Feature selection is an important step for efficient dimensionality reduction to gain quality classifiers [77]. Broadly, three methods for feature selection in use are filter methods, wrapper methods and embedded methods. However, it is hard to determine any one specific method as the most accurate [78]. As the accuracy of machine learning approaches is highly dependent on the selected features [79], we used a combination of data-driven filter methods and other feature selection methods (recursive feature elimination, Gini index, and random forest feature selection) to select the optimal features. Initially, 1096 features for OR1A1 and 1034 features for OR2W1 respectively, were eliminated, using filter methods (Appendix A). Subsequently, three different methods for feature selection were applied to each selected feature dataset. A wrapper method: recursive feature elimination, a filter method: Gini index, and an embedded method: random forest feature selection were used. The top 20 features from each approach were compared. For OR1A1, 13 consensus features were obtained by the three methods, namely VR2_A, AATSC0v, ATSC6d, ATSC7d, ATSC8c, BCUTm.1l, C3SP2, EState_VSA4, JGI5, JGI6, JGI7, PEOE_VSA6, and SdssC. For OR2W1, there were only two consensus features: JGI5 and JGI6, among the three different methods of feature selection. Therefore, the features predicted by three different methods were iteratively applied to finally select the five features for OR2W1 for training the model (Appendix A). This selection of the features was based on the combination of features giving the maximum accuracy. The detailed description of each selected feature is mentioned in Appendix A. The selected combination of five features gave the maximum accuracy for all classification models. RF, SVM and NB classifiers were hyper-parameterized and trained using the selected features. The SVM classifier showed comparable performance to RF classifier for the OR1A1 dataset, while for the OR2W1 dataset, the RF classifier outperforms the other two classifiers (Figure 3). 

The performance of each classifier was validated by 10-fold CV, and testing data. Additionally, the classifiers (or models) were evaluated on the basis of accuracy, specificity, sensitivity, and F1 metrics. Additionally, we carried out two statistical tests to compare the performance of the classifiers (Appendix A). The *p*-value of each classifier was significant (<2 × 10^−16^) for the training dataset for both ORs while for testing dataset, the RF classifier was close second to the SVM classifier for OR1A1 and scored best for OR2W1-test set. The kappa values for RF classifiers outperformed the other two classifiers for both training and testing sets of OR1A1 and of OR2W1. Based on the classifier evaluation scores, RF was selected as the predictive model for screening the compounds downloaded from five different databases.

### 3.3. Putative Ligand Screening through Machine Learning Based Classification

We downloaded 22,938,816 compounds from five different online databases (ZINC, HMDB, ChEBI, COD, and OdorDB). Since the classifiers were computationally trained on specific molecular descriptors for agonists and non-agonists of OR1A1 and OR2W1, they can only classify compounds belonging to similar chemical space. To identify similar chemically spaced compounds, we applied a Tanimoto index value of 0.85 to the entire downloaded dataset. The compound search space was thus reduced to 35,415 for OR1A1 and 27,127 for OR2W1 (Appendix A). The trained RF classifier ranked these chemically similar compounds as agonists and non-agonists, separately, for each receptor. Based on classification probabilities, compounds were ranked as agonists. As the sensitivity and F1 score values of the RF classifier for testing data were below 0.80, we set up a threshold value of prediction probability as 1.0, in order to limit false positive predictions. Generally, the class membership probability has a threshold value of 0.5 [80]. With this threshold value, 67 compounds (three from ChEBI, one from COD, two from HMDB, three from OdorDB, and 58 from ZINC) were predicted as OR1A1 agonists. Independently, 83 compounds (three from ChEBI, none from COD and OdorDB, and 80 from ZINC) were predicted as OR2W1 agonists.

### 3.4. Homology Model and Molecular Docking Analysis

To identify the top candidates for experimental validation, we conducted induced-fit docking of the highly probable compounds based on 3D homology models of OR1A1 and OR2W1. We built the homology models of OR2W1 (UniProtID: Q9Y3N9) and OR1A1 (UniProtID: Q9P1Q5) using the approach described previously [50], with the X-ray crystallographic structure of bovine rhodopsin (PDB ID: 1U19) [51] as a template, and Bio-GATS TM alignments [30]. The experimental structure of an insect OR has recently been resolved at 3.3 Å resolution [9] with an inverted topology to human ORs. The sequence identity (SI) and query coverage (QC) between the experimentally determined structure (PDBID: 7LID) and our target sequences is extremely low (SI for 7LID-OR1A1 pair: 7.0%, QC: 50% and SI for 7LID-OR2W1 pair: 7.16%, QC: 58%). Due to the low resolution of the insect OR structure and extremely low sequence identity and low query coverage with the human ORs under investigation, we did not proceed with the 7LID template. The structures for predicted ligands and benzophenone (experimentally known ligand for both receptors) were downloaded from PubChem [81] and optimized using the ICM package [56] and docked to the homology models for OR1A1 and OR2W1. The binding site for each receptor was selected as a consensus site considering the experimental mutagenesis sites for the other ORs (OR1A1, OR1A2, OR1G1, OR2AG1, OR2M3, OR5AN1, OR7D4 and OR51E2), based on the alignment of OR1A1, OR2W1 and the OR sequences with available mutagenesis data shown in Appendix A. Positions G108^3.35^, S109^3.36^, C112^3.39^, N155^4.56^, I206^5.46^ and Y252^6.48^ of the predicted binding pocket are consistent with the available OR mutagenesis data for OR2W1 (numbering in superscripts are the respective Ballesteros-Weinstein residue numbers [82]). Also, positions G108^3.35^, N109^3.36^, S112^3.39^, I205^5.46^, Y251^6.48^, Y258^6.55^ and T277^7.42^ of the OR1A1 binding pocket are consistent with mutagenesis data available for OR1A1. All predicted ligands and benzophenone were docked to their respective receptor model, with 100 conformations were generated for each predicted ligand and the control (benzophenone). The ICM docking score of benzophenone was used as a threshold to select the docked compounds with equivalent scores. The conformation that fits within the binding pocket and has an ICM docking score around that of benzophenone, was selected. This strategy reduced the predictions to 23 compounds for OR1A1 and 10 compounds for OR2W1, with ICM scores nearest to that of benzophenone (Table 1 and Table 2). Of these, four compounds for OR1A1 and two compounds for OR2W1 that were not reported in the Bushdid et al. [14] study were randomly selected, and experimentally tested using functional in vitro assays.

### 3.5. In Vitro Testing of Predicted Agonists Using Luciferase Assay

We tested the response of OR1A1 and OR2W1 to different concentrations of the candidate molecules using an in vitro luciferase assay (Figure 4A,B), following the verification of cell surface expression of these two ORs by flow cytometry (see Methods). We used Olfr539 and Olfr541 as positive and negative controls of OR cell surface expression, respectively [74]. In comparison to our controls, both ORs are relatively well trafficked to the cell surface (Appendix A).

Benzophenone was added to the set of tested molecules as a positive control for OR1A1 and OR2W1 activation as it is an agonist for both ORs [83]. Ethyl 2-(3-bromophenyl)acetate, methyl 2-(2-methylphenyl)acetate and 1,10-phenanthroline stimulations were tested for OR1A1 activation in dose-responses (Figure 4A). Both ethyl 2-(3-bromophenyl)acetate and methyl 2-(2-methylphenyl)acetate were able to activate OR1A1 and showed similar dose-response curves and EC_50_ values (EC_50_ (ethyl 2-(3-bromophenyl)acetate) = 11.5–24.4 µM; EC_50_ (methyl 2-(2-methylphenyl)acetate) = 12.2–25.0 µM, 95% CI). Dipentene (racemic limonene) was a weak activator, while 1,10-phenanthroline did not activate OR1A1 and was considered a non-agonist. 1,4-bis(4-Vinylphenoxy)butane and 2-ethoxynaphthalene stimulation were tested for OR2W1 (Figure 4B). 2-Ethoxynaphthalene activated OR2W1 in a dose-response manner with an EC_50_ of 6.59–3.05 µM (95% CI). 1,4-bis(4-Vinylphenoxy)butane was identified as a non-agonist of OR2W1.

### 3.6. Binding Mode of the Tested Ligands

The two activating agonists for OR1A1 and the activating agonist for OR2W1 were re-docked in their respective receptor’s binding pocket, to analyse the receptor binding residues and the binding mode for these novel agonists. Interacting residues of the individual receptors are shown in Appendix A. Ethyl 2-(3-bromophenyl)acetate has a single hydrogen bond to S112^3.41^ within the OR1A1 binding pocket, while the rest of the interactions are hydrophobic. Methyl 2-(2-methylphenyl)acetate shows hydrophobic interactions with OR1A1. 2-Ethoxynaphthalene also shows predominantly hydrophobic interactions with OR2W1.

## 4. Discussion

In the current study, we report the ML-based virtual screening workflow for agonist identification of two broadly tuned ectopic ORs: OR1A1 and OR2W1. Both receptors have physiological and pathophysiological implications. In an earlier study, SVM was applied to OR1A1 and OR2W1 to screen a test set of 258 compounds and resulted in the identification of novel agonists for both receptors, with a hit rate of 39 to 40% for these ORs [14]. In this present work, we further build hyper-parameterized RF and NB models along with SVM, selected the best performing RF model and achieved a hit rate of 75% for OR1A1 and 50% for OR2W1, respectively. The dataset, comprised of experimentally known agonists and non-agonists for both ORs, is highly imbalanced. Therefore, careful selection of features as well as the classification model is necessary. Further, the dataset for OR2W1 is extremely diverse, show high variance and low biased as compared to OR1A1 (Figure 2) which indicates that OR2W1 dataset is more prone to overfitting. The right balance between variance and bias is desired, to have an optimal ML model [84]. Therefore, we carefully selected the features to train the classifiers by applying filter-based, wrapper, and embedded methods. We have selected five features to suit the size of the data sets. Moreover, all three models were hyper-parameterized to avoid any overfitting that might occur due to decreased bias. As a result, we obtained models showing reasonably good classification accuracy, both on training and testing data, as shown in Figure 3. We then compared the performance of three well-established ML classifiers based on accuracy, sensitivity, specificity and F1 score. The hyper-parameterized RF classifier outperforms the other classifiers, SVM and NB, for both receptors with all values exceeding 0.85 for training data and thus capable of distinguishing agonists from non-agonists. However, sensitivity and F1 score for testing data was below 0.80. Therefore, we set up a threshold value of prediction probability to 1.0, in order to avoid excessive false positives. Generally, the class membership probability has a threshold value of only 0.5 [80], indicating that our selected threshold is much more stringent. Based on performance, we selected the RF classifier and used it to screen the huge test set of 22,938,816 compounds from five compound databases. Scanning similar spaced test set compounds with the RF classifier yielded 67 and 83 compounds ranked as agonists for OR1A1 and OR2W1, respectively. We docked these compounds into the binding pocket of the respective receptor structural models, to further validate our predictions. Compounds showing good binding affinity in docking runs were shortlisted and randomly selected for experimental testing through luciferase assays, to evaluate the validity of our approach. Of the four compounds tested for their responsiveness against OR1A1, ethyl 2-(3-bromophenyl)acetate and methyl 2-(2-methylphenyl)acetate are identified as high activity novel agonists for OR1A1. We also identified dipentene (racemic limonene), as an activating ligand for OR1A1. The two isomers of limonene that are *(S)*-(-)-limonene and (*R*)-limonene have already been identified as agonists for OR1A1 in multiple studies (Appendix A). 1,10-Phenanthroline did not activate the receptor and therefore, was regarded as a non-agonist for OR1A1. We also tested two compounds for their activity against OR2W1. 2-Ethoxynaphthalene turned to be the agonist, while 1,4 bis(4-vinylphenoxy) butane was regarded as a non-agonist. Our results are consistent with the observations of Bushdid et al. [14], where agonist and non-agonist spaces could not easily be differentiated on the basis of simple chemical descriptors. Evaluating the experimental affinity between ORs and odorant molecules could be helpful to establish a performant predictive model. Unfortunately, there are no such data available today for mammalian ORs.

We further compared the potency of identified agonists with the previously identified potent agonists for OR1A1 and OR2W1 as illustrated in the supporting information of Bushdid et al. [14]. Agonists identified in this study show a triggered response of >80% of the control (benzophenone) for OR1A1, while the agonist identified for OR2W1 shows 90% triggered response of the same control (benzophenone) (Figure 5). Although (−)-carvone is a stronger control for OR1A1 as compared to benzophenone, the novel agonists identified in this study show comparable triggered response to (−)-carvone (>75%). 2-ethoxynephthalene, identified as an agonist for OR2W1 in the current study is two times more potent than the strongest agonist identified for OR2W1 in the Bushdid et al. [14] study. Identification of highly potent agonists demonstrates the efficacy of our classification model.

We re-analysed the putative binding sites of novel agonists with the binding pocket of respective ORs. It is being reported by multiple studies that ligand binding niche for many ORs comprised of TM3, TM5, TM6 and TM7 [59],[85]. The putative interacting sites of all three novel agonists lie within the proposed ligand binding cradle of ORs (Appendix A). Also, the residues G108^3.36^, N109^3.37^, S112^3.40^, I205^5.46^, Y251^6.48^, Y258^6.55^, T277^7.42^ of OR1A1 have already been validated experimentally to be part of ligand binding pocket through site directed mutagenesis [20,21]. OR2W1 does not have any site directed mutagenesis data available yet, but the putative agonist binding residue positions within the receptor have been recognised as important in defining ligand binding cradle for ORs (Appendix A). Position 3.36, 3.37, 5.46 (G108, S109, I206 in OR2W1) are part of ligand binding pocket in OR1A1 and OR1A2 [20]. Position 3.40 (C112 in OR2W1) is an important binding cradle position for OR1A1, OR1A2 [15], OR1G1 [23], and OR51E2 [24] while 6.48 is important for ligand binding in OR1A1. The position 7.42 is crucial for ligand binding in OR1A1 and OR7D4 [86].

In summary, we have identified two high activity agonists for OR1A1 and one high activity agonist for OR2W1 through binary classification based on RF model. The data driven approaches like ML coupled with in vitro approaches are well suited for linking odorants to their respective ORs. The proposed workflow is generic and applicable to other broadly tuned olfactory receptors including OR52D1 and a PSGR i.e., OR51E2 for discovering further high affinity ligands. Unfortunately, the majority of the ORs are either narrowly tuned or orphans so ML methods cannot be applied for discovering agonists for these ORs. Moreover, ML models can only classify the compounds that overlap the chemical space of already known compounds and are limited by their applicability domain. Other methods such as pharmacophore-based virtual screening and structure based virtual screening might be helpful in identifying structurally different agonists.

## Figures and Tables

**Figure 1 ijms-22-11546-f001:**
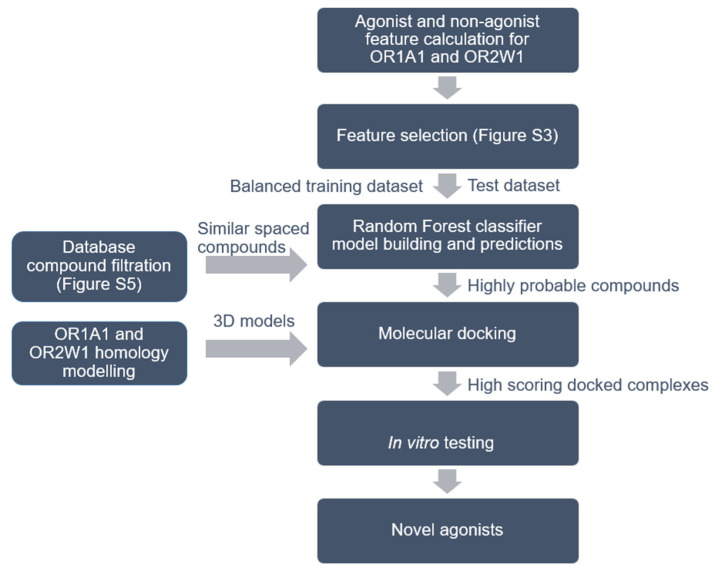
Workflow for agonist identification based on machine learning, molecular docking and in vitro testing. Details of feature selection is shown in Appendix A and the process for database filtration is shown in Appendix A. Compounds from five different databases were downloaded and classified as agonists or non-agonists for OR1A1 and OR2W1.

**Figure 2 ijms-22-11546-f002:**
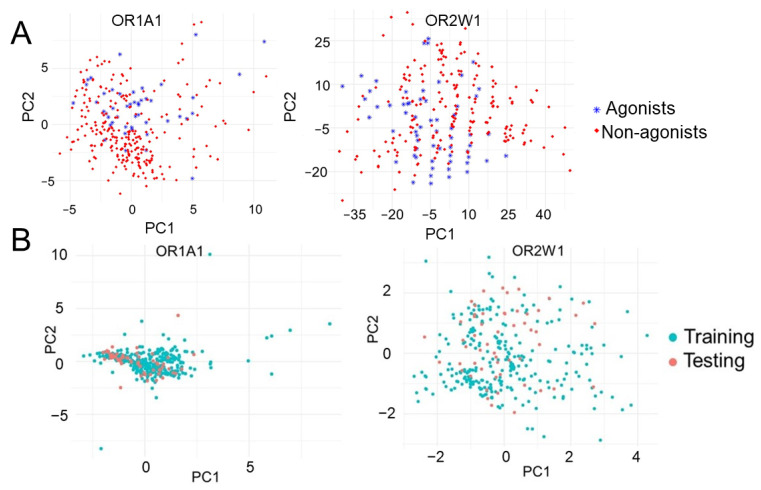
First two principal components of (**A**) OR1A1 and OR2W1 agonists and non-agonists showing the diversified nature of agonists for both ORs and (**B**) training and testing datasets showing the considerable overlap between training and testing set.

**Figure 3 ijms-22-11546-f003:**
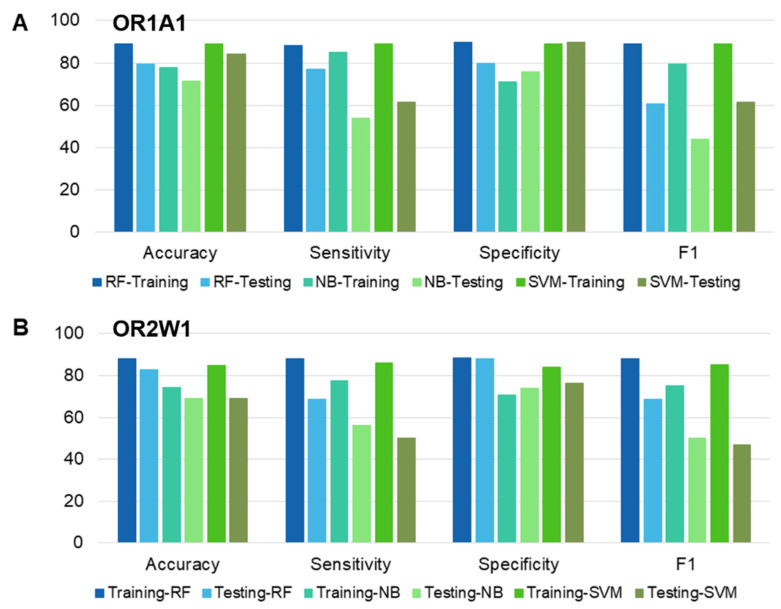
Performance comparison of different classifiers based on accuracy, sensitivity, specificity, and F1 score for (**A**) OR1A1 (**B**) OR2W1. Training and testing datasets for each OR have been compared, using RF: random forest, NB: Naïve Bayes and SVM: Support vector machine.

**Figure 4 ijms-22-11546-f004:**
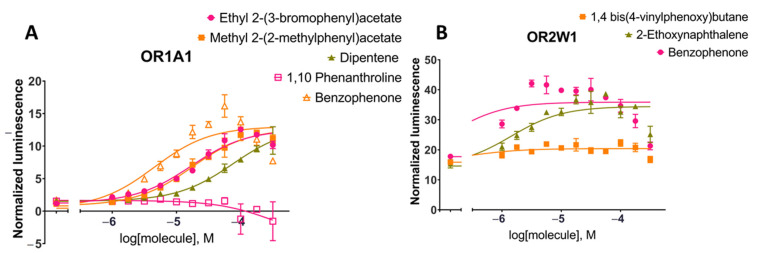
Dose-response curves for tested compounds against (**A**) OR1A1 and (**B**) OR2W1 for the luciferase assay (see Methods). The tested compounds were randomly selected from short-listed compounds after machine learning and molecular docking to evaluate the random forest model predictions. Cell surface expression of these two ORs from flow cytometry are shown in Appendix A.

**Figure 5 ijms-22-11546-f005:**
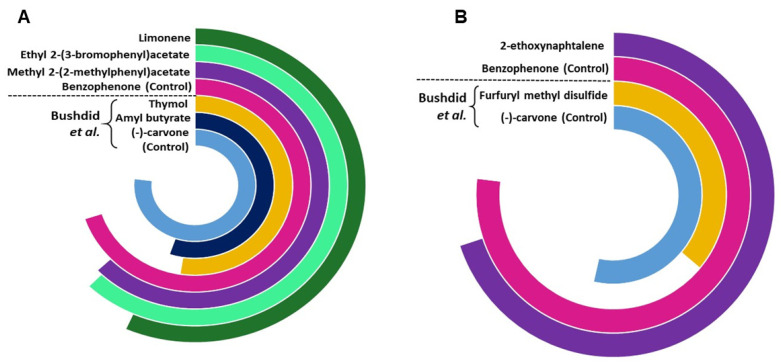
The comparison of potency of agonists and control identified in the current study compared with those reported by Bushdid et al. [14], for (**A**) OR1A1 and (**B**) OR2W1. The agonists reported in the current study are more potent than the agonists previously reported by Bushdid et al. [14] for both ORs.

**Table 1 ijms-22-11546-t001:** Highly probable OR1A1 agonists based on docking scores. Control in italics; experimentally test compounds underlined.

PubChem_CID	Compound Name	Database	Chemical Nature	ICM Docking Score
*3102*	* Benzophenone *	*Control*	*Ketone*	*−12.8345*
10465547	[(*Z*)-Pent-3-enyl] 2-aminobenzoate	ZINC	Heterocyclic compound	−24.058033
70545042	Prop-2-enyl 3-iodobenzoate	ZINC	Heterocyclic compound	−20.113147
56806459	2-(4-Methylphenoxy)pentan-3-one	ZINC	Ketone	−19.297316
84603836	(3-Fluorophenyl)methyl 4-methylpentanoate	ZINC	Ester	−18.084389
101977	D-citronellol	ChEBI, HMDB	Terpene	−18.009
56828593	4-(3-Fluorophenyl)-3-methyl-4-oxobutanenitrile	ZINC	Heterocyclic compound	−17.436507
17973047	4-(1-Methylcyclopropyl)phenol	ZINC	Heterocyclic compound	−17.286264
30842889	Prop-2-enyl 2-(2,4-difluorophenyl)acetate	ZINC	Ester	−17.136729
22048986	6-Chloro-1-(3-fluorophenyl)hexan-1-one	ZINC	Ketone	−17.053948
11470552	Ethyl 2-(3-bromophenyl)acetate	ZINC	Ester	−16.812695
45085600	(5*S*)-5,6-dimethylhept-6-en-2-one	ZINC	Ketone	−16.733556
5352782	3-[(*E*)-But-1-enyl]pyridine	ZINC	Heterocyclic compound	−16.708306
7021479	Methyl 2-(2-methylphenyl)acetate	ZINC	Ester	−16.649985
59382573	(3-Methoxyphenyl)methyl butanoate	ZINC	Ester	−16.572913
84177	Ethyl 2-(4-chlorophenyl)acetate	ZINC	Ester	−16.278326
6368521	1-[(*E*)-2-Chloroethenyl]-4-methoxybenzene	ZINC	Heterocyclic compound	−16.266073
78901972	(3-Fluorophenyl)methyl 2-propylsulfanylacetate	ZINC	Ester	−16.165344
8842	Citronellol	OdorDB, ChEBI	Terpene	−14.6017
22311	Dipentene	OdorDB, COD	Terpene	−13.9064
1318	1,10-Phenanthroline	ChEBI	Heterocyclic compound	−13.7334
24473	Dihydrocarvone	OdorDB	Ketone	−11.3054
131752167	2,10-Bisaboladiene-1,4-diol	HMDB	Alcohol	−10.6224
78236	4-Nonanone	HMDB	Ketone	−10.5291

**Table 2 ijms-22-11546-t002:** Highly probable OR2W1 agonists based on docking scores. Control in italics; experimentally test compounds underlined.

PubChem_CID	Compound Name	Database	Chemical Nature	ICM Docking Score
*3102*	* Benzophenone *	*Control*	*Ketone*	*−11.8875*
13433021	4-Methyl-2-m-tolylpyridine	ZINC	Heterocyclic compound	−14.515694
2733871	2,4-Dimethyl-1-phenylpyrrole	ZINC	Heterocyclic compound	−14.150921
249799	1-Butoxy-4-phenylbenzene	ZINC	Heterocyclic compound	−13.419691
22562335	Methyl 3-(4-ethoxyphenyl)prop-2-ynoate	ZINC	Heterocyclic compound	−12.639271
16530415	(2,3,4,5,6-Pentafluorophenyl)methyl 2-hydroxy-3-methylbenzoate	ZINC	Heterocyclic compound	−11.02916
3847415	1-Ethenyl-4-[4-(4-ethenylphenoxy)butoxy]benzene	ZINC	Heterocyclic compound	−8.767672
12252872	Ethyl 4-hydroxy-3-prop-2-enylbenzoate	ZINC	Heterocyclic compound	−8.68684
231770	1,3-bis(4-Bromophenyl)prop-2-en-1-one	ZINC	Heterocyclic compound	−8.668044
7129	2-Ethoxynaphthalene	ZINC	Ether	−8.208685
60008260	Ethyl 2-amino-5-cyanobenzoate	ZINC	Heterocyclic compound	−8.011839

## Data Availability

The data presented in this study are available in Appendix A.

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
