# Peer review of "Machine Learning Assisted Approach for Finding Novel High Activity Agonists of Human Ectopic Olfactory Receptors"

_ijms, 2021, doi:10.3390/ijms222111546_

Round 1

Reviewer 1 Report

The manuscript proposal title “Machine learning assisted approach for finding novel high activity agonists of human ectopic olfactory receptors” uses interesting bioinformatics approaches to predict agonists for the two olfactory receptors. However, I think authors need to include more biochemical assays to confirm the bioinformatics results. The following comments are based on my professional experience and addressed to help improve your proposal.

Mayor comments:

Which is the percentage of sequence homology among the OR structure reported by Marmol et al. with respect to OR1A1 and OR2W1?

If ORs were first reported in 1991 and are implicated in several pathologies,  there is an additional limitation with it so that there is no experimental structure for any animal OR?

OR are not easily expressed on the cell surface, since it is key for current assays and the cells were tested only after 18-24 hours, the authors did conventional immunostaining, if so, an image would be appreciated in the supplementary material. Otherwise, more information and results about flow cytometry will be appreciated. Dose-response curves without more information in figure legend cannot support that  “Both ORs are well-trafficked to the cell surface (Figure 4A and 4B)”.

The authors used Hana3A and HEK293T cells why? Looks also Cellosaurus PC2 cells? The results could be extrapolated from one line to another.

The authors claim in discussion  ”ethyl 2-(3-bromophenyl)acetate and methyl 2-(2- methylphenyl)acetate are identified as high activity novel agonists for OR1A1” however, only present a Dual-Glo luciferase reporter gene assay. To confirm it that other assays must be done like competition assay showing affinities, cell signaling transduction.

In the all-figure legends, more information must be added.

In the Methods, the authors include results like “We added the surface expression levels of Olfr539, which was robustly expressed on the cell surface, and Olfr541, which showed no detectable cell surface expression as controls”

Minor comments

Check spaces in the summary and through the text.

Author Response

Thank you for your valuable comments. We have now revised the manuscript thoroughly to address your concerns, wherever possible. We have highlighted the revisions in different colours to enable tracking of changes made.

We have added to the revised manuscript several new references, a supplementary figure and a supplementary table, and have updated Figure 2, along with extensive text revisions.

Our point-wise responses are detailed in the attached PDF.

Reviewer 2 Report

In this manuscript, i have a suggestion.

in introduction: the content and the writing of the general part could be modified to review the syntax of some sentences. 

overall that is well organized.

Author Response

(The authors gave the same response as above.)

Reviewer 3 Report

In this study, Jabeen et al. proposed a machine learning approach for finding novel high activity agonists of human ectopic olfactory receptors. The idea looks interesting and the authors could reach a promising performance in the model. I have some major comments as follows:

1. It would be better if the authors could have an external validation data to evaluate the performance on different data.

2. Why did the authors set the threshold of 0.70 in correlation filter? Any rule for p-value or cut-off value?

3. The authors applied wrapper method for feature selection, but it is not a state-of-the-art methods. Therefore, it is suggested that the authors would conduct more advanced feature selection techniques and have a baseline comparison among them.

4. The authors should compare the predictive performance to previous studies on the same problem/data.

5. Machine learning has been used in previous bioinformatics studies i.e., PMID: 33036150, PMID: 34502160. Thus, the authors are suggested to refer to more works in this description to attract a broader readership.

6. Fig. 2A is for training or testing dataset?

7. Statistical tests should be conducted to see significance of the model.

Author Response

(The authors gave the same response as above.)

Round 2

Reviewer 1 Report

Congratulations on the work you are doing. Thanks for the answers and the changes introduced. I hope you continue with the pharmacological characterization of the OR ligands described. All the best.

Reviewer 3 Report

My previous comments have been addressed well.